# Direct observation of frequency modulated transcription in single cells using light activation

Daniel R Larson[1]*, Christoph Fritzsch[2], Liang Sun[3,4], Xiuhau Meng[5], David S Lawrence[3,4], Robert H Singer[5,6]

[1]Laboratory of Receptor Biology and Gene Expression, Center for Cancer Research, National Cancer Institute, National Institutes of Health, Bethesda, United States; [2]Institute of Molecular Biology, Johannes Gutenberg University of Mainz, Mainz, Germany; [3]Department of Chemistry, Division of Chemical Biology and Medicinal Chemistry, University of North Carolina, Chapel Hill, United States; [4]Department of Pharmacology, University of North Carolina, Chapel Hill, United States; [5]Department of Anatomy and Structural Biology, Albert Einstein College of Medicine, New York, United States; [6]Gruss-Lipper Biophotonics Center, Albert Einstein College of Medicine, New York, United States

**Abstract** Single-cell analysis has revealed that transcription is dynamic and stochastic, but tools are lacking that can determine the mechanism operating at a single gene. Here we utilize single-molecule observations of RNA in fixed and living cells to develop a single-cell model of steroid-receptor mediated gene activation. We determine that steroids drive mRNA synthesis by frequency modulation of transcription. This digital behavior in single cells gives rise to the well-known analog dose response across the population. To test this model, we developed a light-activation technology to turn on a single steroid-responsive gene and follow dynamic synthesis of RNA from the activated locus.

*For correspondence: dan.larson@nih.gov

## Introduction

Steroid receptors coordinate a diverse range of responses in higher eukaryotes and are involved in a wide range of human diseases (*Gronemeyer et al., 2004*). Steroid receptor response elements are present throughout the human genome and modulate chromatin remodeling and transcription in both a local and long-range fashion (*John et al., 2011*). As such, steroid receptor-mediated transcription is a paradigm of genetic control in the metazoan nucleus. Moreover, the ligand-dependent nature of these transcription factors makes them appealing targets for therapeutic intervention, necessitating a quantitative understanding of how receptors control output from target genes.

The classic sigmoidal dose response of steroid-regulated gene products belies the complexity of a system which relies on an intricate, multi-step sequence of events to initiate transcription from a target gene (*McKenna et al., 2009*). Depending on the particular steroid receptor, a partial list of events required to activate the target gene includes ligand-binding, receptor dimerization, nuclear translocation, eviction of co-repressors (i.e., histone deacetylases), recruitment of chromatin modifying enzymes (histone acetyltransferases, ATP-dependent remodeling enzymes, methyltransferases), and eventually recruitment of the basal transcription machinery. Despite this complexity, the dose-response is deceptively simple: it often takes the form of a simple Hill function with a coefficient of unity, which has traditionally been interpreted to imply that ligand-binding is the only rate-limiting step in the activation pathway (*Ong et al., 2010*). This description models the dose response as a continuum where each cell transcribes RNA at rates proportional to the dosage level.

**eLife digest** The process by which a gene is expressed as a protein consists of two stages: transcription, which involves the DNA of the gene being copied into messenger RNA (mRNA); and translation, in which the mRNA is used as a template to assemble amino acids into a protein. Transcription and translation are controlled by many interlinked pathways, which ensures that genes are expressed when and where required.

One of these regulatory pathways involves steroid receptors. The binding of a steroid molecule to its receptor causes the receptor to move into the nucleus and interact with a specific gene, triggering transcription of that gene. When measured at the level of the whole organism, this transcriptional response is dose-dependent—the more steroid molecules that are present, the greater the amount of transcription. However, this is not the case in single cells, in which transcription is either activated or not. This 'on/off' behaviour is also seen over time: steroid-activated transcription occurs in bursts, separated by periods of inactivity.

To unravel the molecular mechanism behind this phenomenon, Larson et al. created a light-activated form of the ligand that activates a specific steroid receptor. Using this molecule, they were able to switch transcription of the gene controlled by that receptor on and off. They then used fluorescent proteins to label the mRNA and protein molecules that were produced as a result.

They found that activating the steroid receptor increases the likelihood of transcription occurring inside a cell, but not the duration of individual bursts of transcriptional activity, nor the amount of mRNA produced during each burst. Activation of a steroid receptor seems to control transcription by reducing the length of time each cell spends in the 'off' state between bursts.

Larson et al. incorporated their findings into a model that also takes into account the natural variability in levels of transcription between cells, and found that this could explain how the digital (on/off) control of transcription at the cellular level leads to analogue, dose-dependent control at the level of a whole organism. These findings should lead to further insights into how transcription is controlled at the molecular level.

However, single-cell studies of gene expression demonstrate that this description is incorrect. First, gene expression is not uniform over a population for a given dose, but shows variation from cell to cell, an observation which was first made for a steroid-responsive MMTV reporter gene (*Ko et al., 1990*). The authors demonstrated that the observed analog dose response was in fact a digital dose response (on or off) when viewed in single cells. Likewise, previous studies initially demonstrated that enhancers increase the *probability* of activation of a cell but not the strength of activation in the cell (*Moreau et al., 1981*; *Weintraub, 1988*; *Moon and Ley, 1991*; *Walters et al., 1995*). Second, expression is only a snapshot of temporally evolving gene activity (*Ross et al., 1994*; *White et al., 1995*; *Harper et al., 2011*; *Suter et al., 2011*). Thus, a cell is counted as activated or not activated dependent on the moment it is observed. Thus far, all single-cell measurements on metazoans suggest that genes are transcribed in 'bursts' of transcription, meaning that short periods of RNA synthesis are interspersed by long periods of inactivity. The causes of transcriptional bursting are unknown. Third, since molecules involved in regulating transcription are usually present at low copy number, this leads to stochastic fluctuations ('noise') and hence gene expression variation across the population (*Larson et al., 2009*). Finally, dynamic interactions between upstream regulators and chromatin add another level of complexity to the molecular events occurring during transcriptional activation (*McNally et al., 2000*; *Darzacq et al., 2009*; *Ong et al., 2010*). Under such conditions, the observed dose response does not result solely from ligand-binding but rather the composite result derived from many coupled reactions. In summary, population models of gene activation are too coarse to explain activation in single cells. Moreover, tools do not exist whereby the activity of single genes in single cells can be directly manipulated and measured.

In this work, we describe an approach for activating a steroid-receptor in order to achieve high temporal and spatial precision and to measure the activity of a responsive gene in the same cell over time. In contrast to the ensemble approach derived from observations of cell populations, we have developed a single-molecule kinetic approach for interrogation of a single gene. This approach is based

on photoactivation of a steroid receptor ligand followed by observation of pre-mRNA synthesis at an active locus. The system consists of an exogenous reporter gene under control of the ecdysone receptor which is activated by the agonist ponasterone A (*No et al., 1996*). The real-time behavior of the gene is visualized using a bacteriophage capsid protein which binds MS2 RNA stem loops with high affinity to label nascent pre-mRNA in living cells (*Bertrand et al., 1998*). We demonstrate experimentally how the ensemble steroid dose response arises from the stochastic behavior of individual genes. These results suggest that the response element controls the *frequency* of gene activity but affects neither the *duration* of the active period nor the actual *rate* of transcripts produced during an active period. By using a caged ligand that could be uncaged by a light pulse (*Lee et al., 2009*; *Lin et al., 2002*) we measured the impulse-response of the gene and determined that a single pulse of active ligand resulted in a corresponding burst of polymerase activity several hours later. Further, this photoactivatable ligand has the property of being an *antagonist* in the caged state and an *agonist* in the uncaged state, enabling a precise window for kinetic perturbation in single cells. Thus, we were able to propose and validate a stochastic model of steroid-receptor activity for a reporter gene which provides a new framework for studying this ubiquitous mechanism of eukaryotic gene regulation.

## Results

We sought to design a reporter system that would enable visualization of multiple steps in gene expression: 1) the nascent pre-mRNA as it is synthesized by RNA polymerase II at an activated locus, 2) the completed mRNA in the cytoplasm and 3) the protein product produced from the mRNA. Furthermore, we desired a gene that would be unresponsive to endogenous steroid receptor (SR) ligands but also recapitulate the basic mechanism of SR transcriptional regulation. For these reasons, we chose to reconstitute ecdysone receptor-mediated transcription in human U2-OS cells (*No et al., 1996*). We first introduced a chimeric ecdysone receptor into the cells and subcloned a cell line which showed a strong response of a luciferase reporter. We next introduced an ecdysone-responsive reporter gene. The gene consists of a multimerized E/GRE response element (hybrid ecdysone/glucorticoid response element) in front of an Sp1 activator and a minimal heat shock promoter (mHSP, *Figure 1A*) (*No et al., 1996*). The coding region is bi-cistronic with the upstream region coding for CFP with a C-terminal SKL amino acid peroxisome-targeting motif (*Janicki et al., 2004*), and the downstream region coding for DsRED expressed in the cytosol from an IRES. Located in the 3′ UTR of the upstream cistron are 24 MS2 stem loops. The entire construct is 7.1 kb in length and is stably infected into the target U2-OS cells using lentiviral vectors (*Mostoslavsky et al., 2006*). Infections were carried out at MOI = 0.1 to obtain cells enriched for single integrations, and multiple clones were isolated and expanded for further analysis. The MS2 RNA stem loops are bound by the high affinity bacteriophage MS2 capsid protein dimer, where each monomer is labeled with YFP (*Figure 1B*) (*Bertrand et al., 1998*; *Chao et al., 2008*). Since the MS2-YFP is constitutively expressed and localized to the nucleus with an NLS, as soon as the RNA stem-loops are transcribed at an active locus, the MS2-YFP protein binds, thus making the transcription site (TS) visible in the microscope (*Figure 1C*, white arrow). The mature mRNAs carry the bound fluorescent proteins into the cytoplasm and are visible in the microscope as punctuate diffraction-limited fluorescent spots (*Figure 1C*, green, inset) (*Shav-Tal et al., 2004*; *Mor et al., 2010*). The translated protein is targeted to peroxisomes which are likewise visible as punctuate diffraction-limited fluorescent spots consisting of many copies of the CFP-SKL (*Figure 1C*, blue). Therefore, the reporter gene provides a live cell fluorescence readout that reflects both the history of the gene (total amount of mRNA and protein) and the current state of the gene (presence of the active transcription site).

Alternatively, these cells when fixed can also be used for fluorescence in situ hybridization (FISH). In this case, we hybridize a fluorescently labeled 20-mer oligodeoxynucleotide to the region between each MS2 RNA stem loop (*Figure 1D*) (*Femino et al., 1998*; *Zenklusen et al., 2008*). FISH is a complementary approach to the live-cell MS2 system and likewise enables direct observation of both nascent RNA at an active transcription site (*Figure 1E*, white arrow) and total cellular reporter mRNA (*Figure 1E*, inset). The benefit of FISH is that it allows very high signal to noise observation of mRNA in fixed cells, while the advantage of the MS2 system is that one can directly observe dynamics in living cells. Both approaches utilize the polymeric nature of mRNA to label single molecules with large numbers (>40) of fluorophores, and both approaches enable quantification of the earliest steps in gene expression on single molecules, single genes and in single cells (*Figure 1F*) (*Larson et al., 2009*).

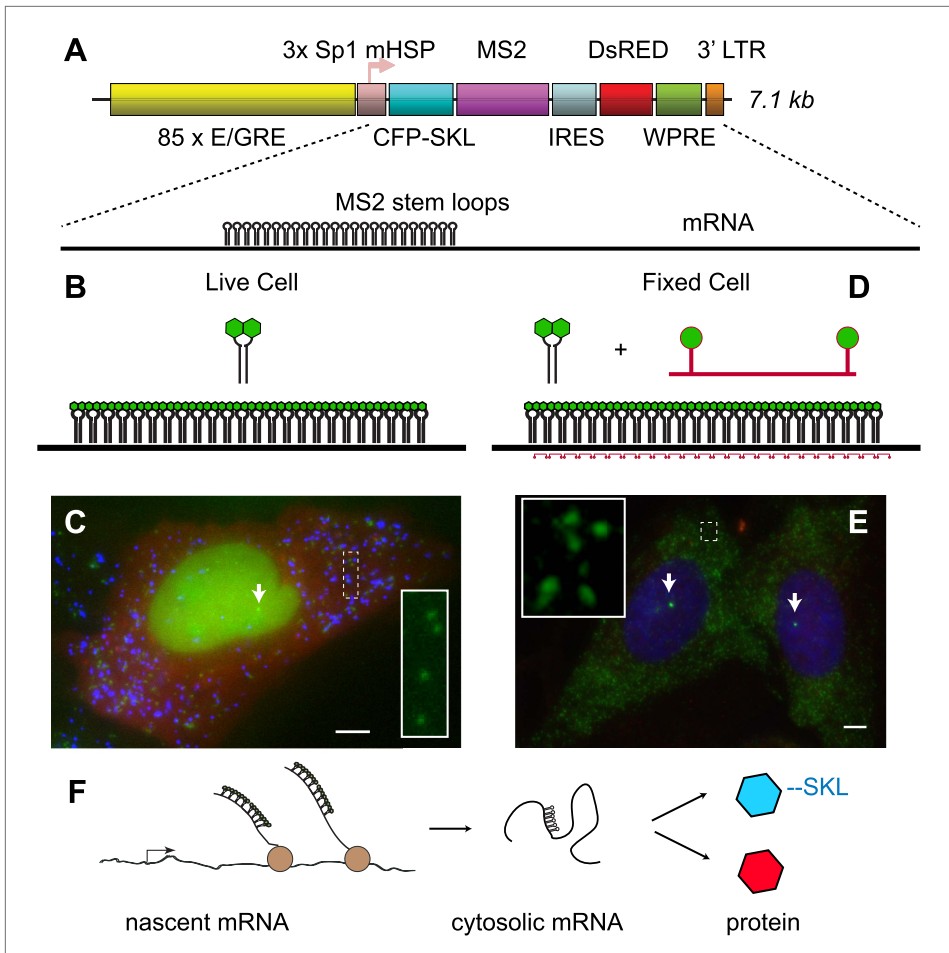

**Figure 1**. Reporter gene design enables observation of multiple steps in gene expression. (**A**) The reporter gene consists of a SR response element multimerized 85 ×, upstream of the transcription start site. The gene is a bi-cistronic construct introduced via lentivirus infection. The first ORF codes for CFP-SKL which results in CFP-labeled peroxisomes, and the second ORF codes for DsRED which is non-specifically expressed in the cytosol. In the 3' UTR of the CFP-SKL cistron, there is a 24 × MS2 stem-loop encoding region. mRNA is drawn to scale. Sequence elements: Sp1 = specificity protein one binding site; mHSP = minimal heat shock promoter; LTR = long terminal repeat; WPRE = woodchuck hepatitis virus posttranscriptional regulatory element. (**B** and **C**) Visualization of mRNA and protein in living cells. The MS2 RNA stem loops bind MS2 coat protein which is constitutively expressed, localized to the nucleus with a nuclear localization signal, and labeled with YFP (green). After 24 hr of activation, the peroxisomes appear as punctuate spots (blue) and the DsRED as a diffuse background (red). Individual mRNA in the cytosol appears as diffraction-limited puncta (inset), and the site of active transcription appears a bright fluorescent spot in the nucleus (white arrow). (**D** and **E**) Visualization of mRNA in fixed cells with FISH. 20-mer DNA oligos with two Cy 3 fluorophores are targeted to the inter-stem loop region of the MS2 cassette. Individual mRNA appears as diffraction limited spots (green, inset), and nascent mRNA appears as bright spots in the nucleus (white arrows, nucleus designated in blue by DAPI staining). Scale bar = 4 µm. (**F**) Visualization of the central dogma. The combined approach allows us to image each step of gene expression: from nascent mRNA at the transcription site, to cytosolic mRNA, to protein.

## Direct measurement of transcriptional bursting on single genes in human cells

First, we directly observed RNA synthesis at an active locus in a clonal cell line using single-molecule live-cell microscopy. Consistent with previous studies which directly observe transcriptional activity of single genes in eukaryotes we observed transcriptional bursting (*Chubb et al., 2006*; *Stevense et al., 2010*; *Yunger et al., 2010*). We then measured the bursting dynamics in the microscope for three different concentrations of the steroid Ponasterone A (PA) (3.125 µM, 12.5 µM, 50 µM, control: 0.0 µM), which is the high affinity ligand for the chimeric ecdysone receptor. An example time series is shown

for 50 µM PA (*Figure 2A–F*, *Videos 1–5*). The first transcription sites were visible several hours after induction (*Figure 2—figure supplement 1*). At steady state (24 hr after induction) for each dose, single gene transcription sites were captured at 15 min intervals for ~ 15 hr, resulting in representative transcription intensity traces shown in *Figure 2E,F* (red, green lines, *Video 4*). As is evident from the images and the intensity analysis, individual cells show punctuated periods of transcriptional activity. The intensity traces were fit using a two-state hidden Markov model (*Figure 2E,F*, black lines, 50 µM; *Figure 2—figure supplement 2*, 12.5 µM, 3.125 µM) (*Lee, 2009*), allowing the distribution of on-times and off times to be determined (*Figure 2G,H*). We observed that the duration of inactivity (off-time) decreased in a dose-dependent manner (*Figure 2I*, gray bars). However, the duration of transcription activity (the on-time) did not significantly change over the dose response curve (*Figure 2I*, black bars). Moreover, the average peak brightness did also not change significantly with [PA] (*Figure 2J*). In the case of treatment with a vehicle ([PonA] = 0.0 µM), occasional spurious spots are detected by the tracking algorithm (*Figure 2I*, 0.0 µM, black bar), and the off-time is statistically indistinguishable from the duration of the entire time-lapse video (*Fig. 2I*, 0.0 µM, gray bar, compared to red-dashed line). Since the brightness of the nascent RNA signal reflects the number of individual RNA at the site of transcription (*Zenklusen et al., 2008*; *Larson et al., 2011*) (*Figure 1F*), these data indicate the average number of polymerases which fire during an active period is insensitive to [PA]. This live-cell analysis was also repeated for an additional clone with equivalent results (*Figure 2—figure supplement 3* and below). In summary, direct measurements of transcription activity suggest that changing steroid levels affects the frequency of active periods but neither the duration of the active period nor the number of polymerases fired during the active period. Even for this highly-induced artificial reporter gene, the gene is active approximately 50% of the time at saturation (average on time = 38 ± 6 min; average off time = 35 ± 4 min) and shows a total gene cycling time of approximately 80 min.

We then repeated this experiment on the same plasmid that was transiently transfected instead of genomically integrated by lentivirus (*Figure 3*, *Videos 6 and 7*). The transfected template, which is present at many copies in the nucleus and which is not integrated into chromatin, shows an extremely rapid induction (<20 min) and robust transcription throughout the time series. We were unable to detect transcriptional bursting. These data indicate that chromatin integration is a primary determinant of transcription kinetics for the hormone response, as has been suggested in previous studies (*Archer et al., 1992*).

## Abundance and variation of cellular mRNA indicates that nuclear receptors are frequency modulators

Using the lentivirus approach results in quasi-random insertion of the transgene throughout the genome. Since the kinetics of transcription depend strongly on transgene insertion (*Figure 3*), we sub-cloned three different lines, measured the dose response to [PA], and fit these data to a three-parameter Hill equation for the steroid response (*Figure 4A*):

$$ activity = \frac{A_{max}[PA]^h}{EC_{50}+[PA]^h} \qquad (1) $$

where $A_{max}$ is the amount of reporter mRNA present at saturation, *[PA]* is the concentration of ligand, $EC_{50}$ is the half-maximal response, and *h* is the Hill coefficient (*Ong et al., 2010*). Although the saturation value for the number of mRNA/cell differed substantially for each insertion site (*Figure 4A*, 960 ± 60 mRNA/cell, 430 ± 50 mRNA/cell, 180 ± 10 mRNA/cell, blue, green, red curves respectively), the Hill coefficient did not change significantly (1.2 ± 0.2, 1.3 ± 0.3, 1.2 ± 0.2), nor did the $EC_{50}$ (8.0 ± 1.4 µM, 10 ± 3.2 µM, 9.1 ± 1.5 µM). We also compared the protein dose response to the RNA dose response for one of the cell lines and observed similar behavior (*Figure 4B*). Since the population dose response for this reporter gene is statistically indistinguishable from a first-order Hill response, we also show a fit to a two-parameter Hill equation with the Hill coefficient fixed at unity (*Figure 4B*, black line, $A_{max}$ = 190 ± 10 mRNA, $EC_{50}$ = 9.6 ± 2.0 µM). In summary, the population dose response is well-described by a first-order Hill response function where the Hill coefficient and the $EC_{50}$ are invariant with insertion site, but the saturation level ($A_{max}$) shows fivefold variation among different clones.

We next sought to build and test a quantitative kinetic model capable of elucidating how the classical analog dose response in a population (*Figure 4A,B*) relates to the time-dependent digital behavior observed directly in single cells (*Figure 2*). We used the 'Random Telegraph model', which is a paradigm

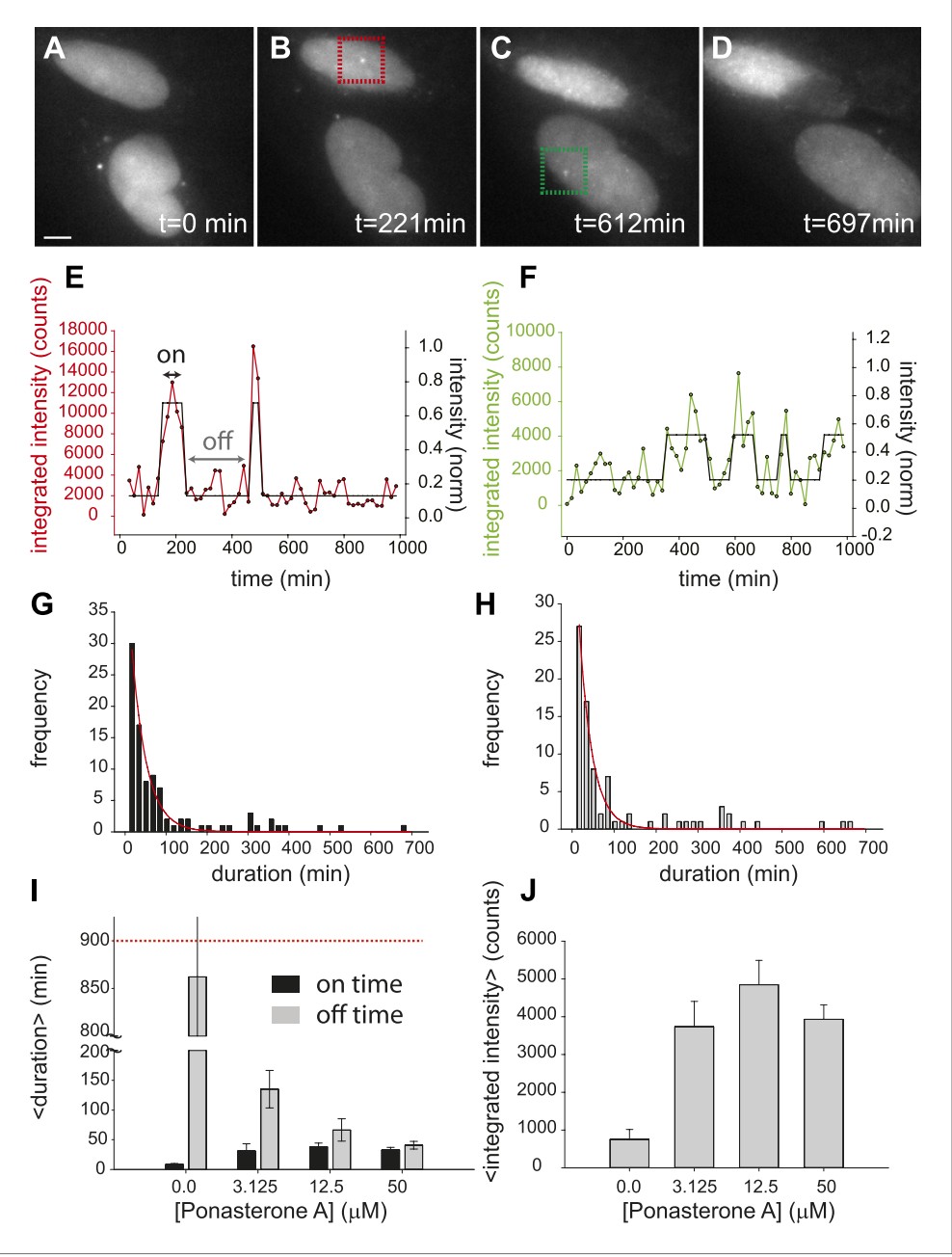

**Figure 2**. Dynamic observation of integrated reporter genes shows transcriptional bursting. (**A–D**) Time-lapse images of active transcription in two separate nuclei. TS are visualized as punctate fluorescent spots where MS2-YFP has coalesced onto multiple MS2-binding sites in nascent pre-mRNA. Scale bar = 4 μm. (**E** and **F**) Quantification of TS intensity for the upper cell/TS (red) and lower cell/TS (green), respectively. The left axis is the integrated intensity of the TS; the right axis is the normalized intensity which is used to fit the intensity trace to a hidden Markov model (black lines) (**Lee, 2009**). Examples of on and off states are designated by the arrows on panel **E**. (**G** and **H**) Histogram of on and off times, respectively. The red lines are exponential fits to the experimental distribution with decay time of 36 ± 6 min and 35 ± 4 min in panels **G** and **H**, respectively (N = 40 cells). [PA] = 50 μM, panels **A–H**. (**I**) Average on (black) and off (gray) duration for four different [PA] obtained from fitting experimental data to an exponential distribution (**Figure 2—figure supplement 2**). The red dashed line indicates the duration of the experiment, which sets the upper limit for off-time values. (**J**) Average TS intensity for four different [PA] (N = 40 cells for each [PA]). Error bars are the SEM.

*Figure 2. Continued on next page*

*Figure 2. Continued*

The following figure supplements are available for figure 2:

**Figure supplement 1**. Time-dependent induction of reporter gene.

**Figure supplement 2**. Dynamic observation of integrated reporter genes shows transcriptional bursting.

**Figure supplement 3**. Dose dependence of on- and off- times for an alternative single cell clone.

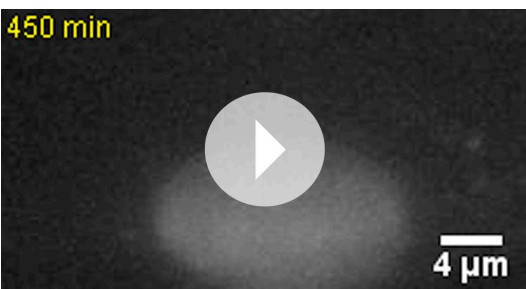

**Video 1**. Time-lapse sequence of reporter gene response to 50 μM PA shows a single pulse of transcription. The nascent transcription site is visualized as the coalescence of MS2-YFP coat protein on newly synthesized pre-mRNA. Each frame is the maximum projection of 18 z-steps acquired at 0.5 μm intervals. Exposure time = 400 ms. Frame interval: 15 min. Total duration: 15 hr. T = 37°C.

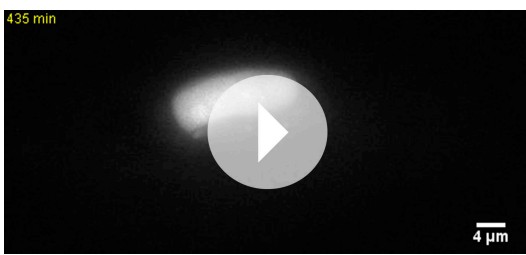

**Video 2**. Time-lapse sequence of reporter gene response to 50 μM PA shows a gene which is on for almost the entire duration of the video. The nascent transcription site is visualized as the coalescence of MS2-YFP coat protein on newly synthesized pre-mRNA. Each frame is the maximum projection of 18 z-steps acquired at 0.5 μm intervals. Exposure time = 400 ms. Frame interval: 15 min. Total duration: 15 hr. T = 37°C.

for stochastic transcription and quantitatively describes the distribution of RNA/cell in a variety of organisms (*Peccoud and Ycart, 1995*; *Larson et al., 2009*). The key feature of this model is the hypothesis that the gene toggles between an active state where transcripts can be synthesized and an inactive state where no transcripts are produced. It is important to note that the gene can be 'active' even when no nascent RNA is detected, owing partly to the stochastic initiation events within an active state (*Figure 4C*) and partly to the placement of the MS2 cassette in the middle of the gene (*Figure 1A*). The biochemical determinant(s) of this active state is unknown and is likely to vary between genes. Consequently, it is not possible at present to do a direct measurement of the active state. However, the distribution of *total* cellular reporter mRNA reflects the dynamics of this active-inactive switching, so we measured the steady state distribution of total cellular reporter mRNA over a population using single-molecule FISH in fixed cells.

For each cell line, the variation of total cellular mRNA within a population is indicative of infrequent bursts of transcriptional activity (*Golding et al., 2005*; *Chubb et al., 2006*; *Raj et al., 2006*; *Voss et al., 2006*; *Zenklusen et al., 2008*) (*Figure 4D–F*). At low PA induction levels (3.12 μM, *Figure 4D*), there is a large heterogeneity of cellular mRNA in the population: some cells contain dozens of mRNAs and others have none. Transcription sites are infrequent. At medium induction levels (12.5 μM, *Figure 4E*), most cells contain mRNA, and many are actively transcribing. At high levels of induction (50 μM, *Figure 4F*), cells contain a high abundance of mRNA in the cytosol, and most cells show an active transcription site in the nucleus. At each concentration of the steroid, we fit the full distribution of mRNA/cell to the steady-state solution for the random telegraph model (*Figure 4G–L*, N = 60 cells at each dose). The steady state distribution function used for fitting is a function of three unitless kinetic parameters (*a/d, b/d, c/d*) corresponding to the rate of activation (a), the rate of inactivation (b), and the rate of firing once the gene is active (c), each normalized by the RNA decay rate, *d* (*Figure 4C*). These parameters correspond to the frequency of bursts, the duration of bursts, and the initiation rate from a burst, respectively. At steady state, the mean level of RNA ⟨N⟩ is:

$$\langle N \rangle = \frac{c}{d}\left(\frac{a}{a+b}\right) \qquad (2)$$

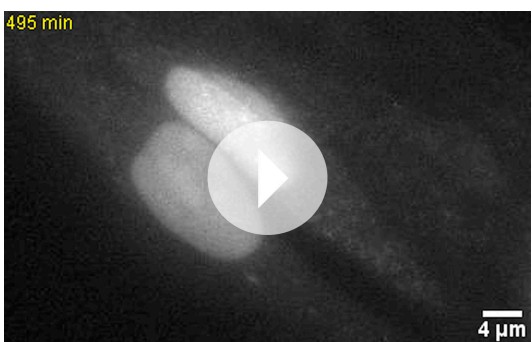

**Video 3**. Time-lapse sequence of reporter gene response to 50 µM PA for multiple cells showing uncorrelated transcription dynamics. The nascent transcription site is visualized as the coalescence of MS2-YFP coat protein on newly synthesized pre-mRNA. Each frame is the maximum projection of 18 z-steps acquired at 0.5 µm intervals. Exposure time = 400 ms. Frame interval: 15 min. Total duration: 15 hr. T=37°C.

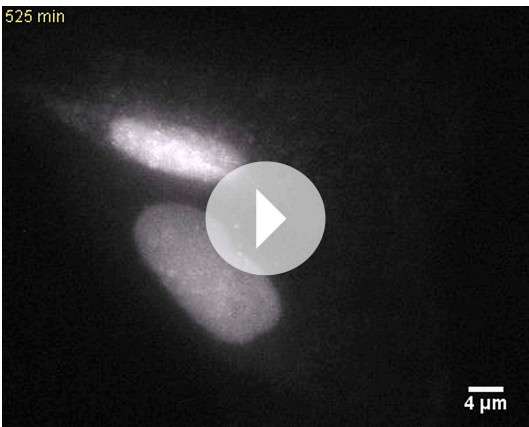

**Video 4**. Time-lapse sequence of reporter gene response to 50 µM PA for cells in *Figure 2*. The nascent transcription site is visualized as the coalescence of MS2-YFP coat protein on newly synthesized pre-mRNA. Each frame is the maximum projection of 18 z-steps acquired at 0.5 µm intervals. Exposure time = 400 ms. Frame interval: 15 min. Total duration: 15 hr. T = 37°C.

where $a/(a+b)$ corresponds to the fraction of time the gene spends in the active state.

There is a functional similarity between the stochastic random telegraph equation (*Equation 2*) and the deterministic Hill equation (*Equation 1*): $A_{max}$ corresponds to $c/d$, $EC_{50}$ corresponds to $b$, and $[PA]$ corresponds to $a$. This correspondence indicates that non-cooperative behavior for the hormone response ($h = 1$, *Equation 1*) is mathematically equivalent to frequency modulation. In other words, the rate of switching to the active state ($a$) is the only variable which changes with ligand concentration. In *Figure 4G–L* we show the distribution of mRNA/cell compared to the theoretical fit arising from the telegraph model as a function of dose. Indeed, these data are consistent with a model where the only variable which varies with dose is the rate of activation ($a$), while the other parameters ($b,c$) are globally fit for the entire dose response for a particular cell line ($b/d = 1.5$, $c/d =340$). When the same analysis is carried out on cell lines with different insertion sites (i.e., *Figure 4A*), the dose-dependent activation rate ($a$) and the dose-independent inactivation rate ($b$) remain invariant, but the initiation rate ($c$) changes with the cell line. Thus, the absolute number of polymerases loaded during an active period does not depend on steroid level but rather some extrinsic factor such as local chromatin environment or abundance of some component of the core transcriptional machinery. This is evident when comparing the episomal genes, which do not show frequency modulation. In summary, both fixed-cell and live-cell single-molecule measurements are consistent with a model where the steroid is a robust frequency modulator.

## Triggering a single activation event in living cells with UV photolysis

We emphasize that the frequency change observed here is in response to changes in the time-invariant level of steroid. Conversely, there is a growing body of evidence that the dynamics of the upstream activator itself may in fact carry biological information and modulate downstream expression levels (*Cai et al., 2008*; *Ashall et al., 2009*; *Stavreva et al., 2009*; *Purvis et al., 2012*). To investigate the temporal relationship between steroid availability and the kinetics of transcription, we required a ligand that could be activated at a precise time to induce transcription. We utilized a caged PA that consists of PA modified at a critical OH residue with a photolabile organic ligand (*Figure 5A*) (*Lin et al., 2002*).

We observed empirically that we were unable to activate single genes in the presence of the caged reagent and hypothesized that the caged PA possessed antagonist properties. We therefore examined the dose-dependent induction properties of the reporter gene in the presence and absence of the caged reagent by performing an in vivo competition experiment. First, we confirmed that both forms of the caged PA were inactive by measuring a dose response of the caged ligands

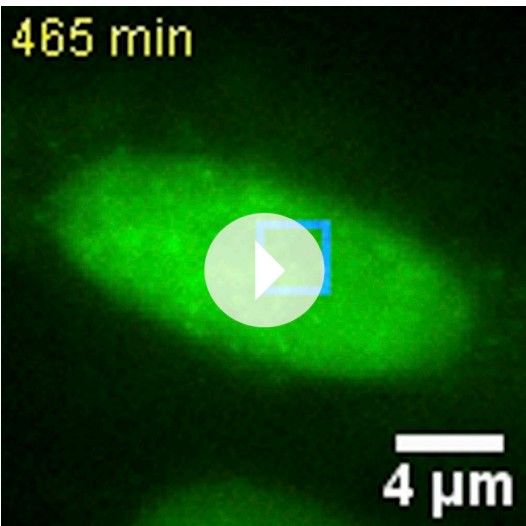

465 min

4 µm

**Video 5**. Post-processing time-lapse sequence of reporter gene response to 50 µM PA for cells in *Figure 2*. The nascent transcription site is visualized as the coalescence of MS2-YFP coat protein on newly synthesized pre-mRNA. Each cell is individually segmented and re-cropped to a separate image file, resulting in a time-lapse sequence where the cell is always centered in the frame. Each frame is the maximum projection of 18 z-steps acquired at 0.5 µm intervals. Exposure time = 400 ms. Frame interval: 15 min. Total duration: 15 hr. T = 37°C.

(*Figure 5—figure supplement 1*). Second, the cells were incubated with two different forms of the caged PA (DMNB-PonA and CNB-PonA) (*Figure 5A*). After overnight incubation, the cells were exposed to 25 µM of the active, unmodified PA for 30 min. The cells were then washed and returned to the incubator for 8 hr to allow the buildup of protein (*Figure 5B*). At 100 µM of DMNB-PA, we observed a fivefold reduction in protein levels, indicating that the caged ligand blocks the activity of the active, uncaged form in vivo (*Figure 5C*, gray, *Figure 5—figure supplement 2*). When the DMNB-PonA and uncaged PA are present in equimolar amounts, the competition is alleviated, and the protein output from the reporter gene is close to the levels observed in the absence of the caged ligand. Thus, we show that this ligand has the unique property of being an *agonist* in the uncaged state and an *antagonist* in the caged state.

The benefit of a reagent that can be switched from repressive to activating is that the effect of small amounts of the agonist are suppressed by the presence of the antagonist. Such a feature is especially important for a reagent that is membrane-permeable and might diffuse to neighboring cells in the tissue. Thus, the caged compound has built-in protection against 'leakiness' and is capable of acting in a highly localized spatial manner when uncaged. The method of activation, therefore, has to be all or none: the balance between agonist/antagonist must be shifted past a threshold in order for gene activation to occur. A typical experimental design is schematized in *Figure 6A*. The nucleus is activated with a UV laser at 349 nm. The above-threshold dose of uncaging light (40 pulses, 100 Hz, 2 J/cm$^2$ total dose at the sample) is delivered in approximately 400 ms over the area of the nucleus (~80 µm$^2$). Now, all the receptors in the nucleus are active, leading to the initiation of transcription at the single locus. Concurrently, any activating ligand that diffuses away encounters neighboring cells that are in the repressed state due to the presence of the antagonist. Once the activation occurs in the target cell,

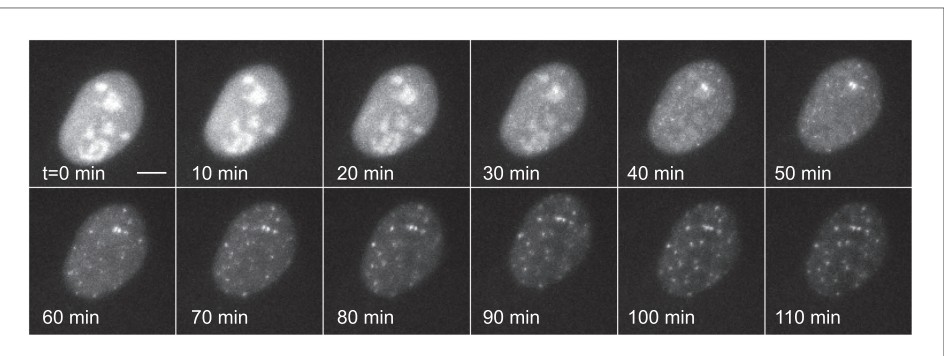

t=0 min | 10 min | 20 min | 30 min | 40 min | 50 min
60 min | 70 min | 80 min | 90 min | 100 min | 110 min

**Figure 3**. Dynamic observation of transiently transfected reporter genes shows fast induction and no bursting. Time-lapse images of transiently transfected cells induced with 50 µM PA at t = 0 min. Each cell contains multiple copies of the reporter plasmid, visible as diffraction limited spots (*Video 6*). Scale bar = 4 µm.

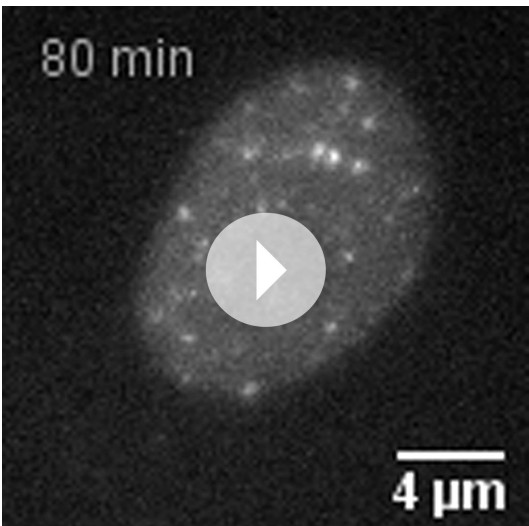

**Video 6**. Time-lapse sequence of a transfected reporter. Multiple nascent transcription sites are visualized as the coalescence of MS2-YFP coat protein on newly synthesized pre-mRNA emerging from the plasmids. Each frame is a single z-plane. Exposure time = 200 ms. Frame interval: 10 min. Total duration: 2 hr. T = 37°C.

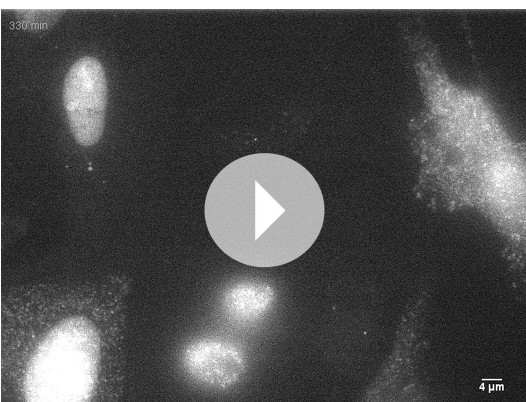

**Video 7**. Time-lapse sequence of a transfected reporter. Multiple nascent transcription sites are visualized as the coalescence of MS2-YFP coat protein on newly synthesized pre-mRNA emerging from the plasmids. Each frame is a single z-plane. Exposure time = 200 ms. Frame interval: 30 min. T = 37°C.

the diffusion into the cell of unactivated antagonist coupled with the rapid turnover of receptor-bound agonist (*Stavreva et al., 2004*) results in antagonist-bound receptors that bind to the response element and repress transcription.

The time evolution of a typical activated cell is shown in *Figure 6* (*Video 8*). The cell is uncaged at *t = 0* (*Figure 6B*); the transcription site is visible at 2.5 hr (*Figure 6C*) and persists in the on state for 1.5 hr afterwards (*Figure 6D*), consistent with the duration of the on-state measured previously (*Figure 2G*). At 15 hr, the CFP peroxisomes are observed to assay whether the cell has activated (*Figure 6E*). In this example, only the central cell was photolyzed, and is the only cell that displays expression of the CFP protein construct. By contrast, the two non-photolyzed neighboring cells show no evidence of CFP expression. The photolysis protocol above results in: 1) an average 106 +/− 36 min between uncaging and the appearance of a transcription site which then persisted for 35 +/− 12 min, 2) 40% success rate for protein expression, and 3) a single on-state in 70% of those cells which activated, with the remainder showing either no detectable on-period (13%) or multiple periods (17%) (N = 30). Moreover, we also observe cell division after uncaging, which is a measure of viability after UV photolysis (*Video 9*). In sum, the photoactivation data suggest that a single pulse of ligand results in a single active period of transcription.

## Discussion

The work presented here addresses the specific question as to how transcription factor availability translates into transcriptional activity. By a variety of approaches, we show that the frequency of initiation is the variable that determines RNA levels, but only when the gene is integrated into the chromatin. By measuring the synthesis of pre-mRNA directly, we connected the dynamics of the gene with the availability of an upstream transcription factor, the ligand-bound steroid receptor. We demonstrated how steroid receptors, which are present in all metazoans, can control transcription through frequency modulation. This digital frequency modulation of transcription in single cells manifests as an analog dose response in cellular mRNA and protein in a population. Importantly, the ability to activate the transcription factor (steroid receptor) using light provided a precise temporal start to measure the time needed for the gene to respond, an unexpectedly long time (2 hr). Hence our observations clarify the mechanics behind these observations and extend the understanding to the intrinsic cycling on and off for a gene. One dose of a transcription factor yields one cycle of transcription.

The precise timing of the transcriptional response places constraints on the regulation of the gene: one can infer regulatory principles based on the observed synthesis of pre-mRNA (*Pedraza and Paulsson, 2008*). A stochastic model of kinetic rates allowed us to break down the steroid transcriptional response

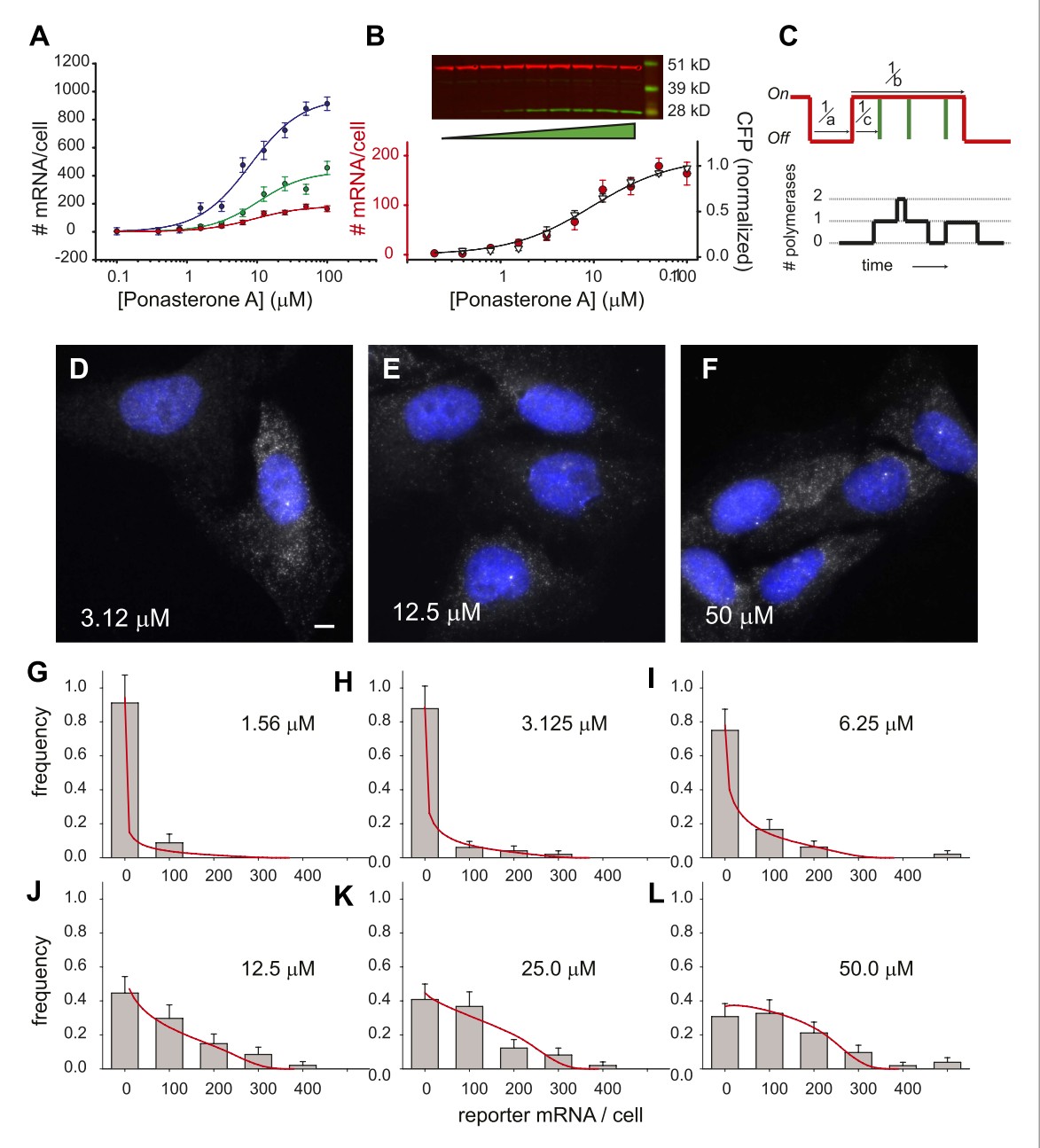

**Figure 4**. Steady state distribution of mRNA from the steroid-activated reporter gene indicates frequency modulation of transcription. (**A**) Dose response of the reporter gene. Total cellular reporter mRNA is plotted as a function of [PA] for three different clones isolated from a population of lentiviral-infected cells (clone 1, red; clone 2, green; clone 3, blue). Each data point is the average value of mRNA/cell determined by automated image segmentation and spot counting (N = 60 cells, error bars are SEM). The fits are three-parameter Hill functions: $A_{max}$ = 960 ± 60 mRNA/cell, 430 ± 50 mRNA/cell, 180 ± 10 mRNA/cell; $h$ = 1.2 ± 0.2, 1.3 ± 0.3, 1.2 ± 0.2; $EC_{50}$ = 8.0 ± 1.4 μM, 10 ± 3.2 μM, 9.1 ±1.5 μM, blue, green, red curves respectively. (**B**) Dose response of mRNA and protein as function of PA concentration for clone 1. The left axis (red circles) is the number of mRNA counted per cell in the microscope as in panel **A**. The error bars are the SEM for N = 60 cells for each concentration. The right axis is the normalized protein level, determined from the quantification of the Western blot shown above. α-Tubulin is used as the loading control. The data are fit with a two-parameter Hill function with h = 1.0 (black line): $A_{max}$ = 190 ± 10 mRNA; $EC_{50}$ = 9.6 ± 2.0 μM. (**C**) Stochastic model of gene induction and the resulting polymerase density. In the random telegraph model (upper panel), the gene exists in an inactive state which is non-permissive to transcription or an active state from which transcripts are produced (on-state indicated by red line). The rate of transition to the active state is $a$; the rate of transition to the inactive state is $b$; the rate of initiation from the active state is $c$. Individual initiation events are indicated by vertical green lines. Lower panel: the RNA polymerase II loading that would result from the telegraph process shown in the upper panel. Each initiation event results in the loading of a polymerase, and that polymerase will have a dwell

*Figure 4. Continued on next page*

*Figure 4. Continued*

time determined by the time necessary to synthesize the nascent transcript. Note that even when the gene is in the active state, it is possible that no polymerases are present (comparing black occupancy trace with red gene activity trace). (**D**–**F**) Fluorescence in situ hybridization at 3.125, 12.5, and 50 µM induction with PA. Gray = Cy3 oligos; blue = DAPI. Scale bar = 4 µm. (**G**–**L**) The steady-state distribution of mRNA/cell for 1.56 µM, 3.125 µM, 6.25 µM, 12.5 µM, 25.0 µM, and 50 µM, respectively. The bin size of the histogram is 100 mRNA. The theory is the full-solution to the Master Equation for the scenario shown in panel **C** ('Materials and methods'). The on rate *a* determined from the fit varies with the dose, but the parameters *b/d* and *cd* are kept constant at 1.5 and 340, respectively.

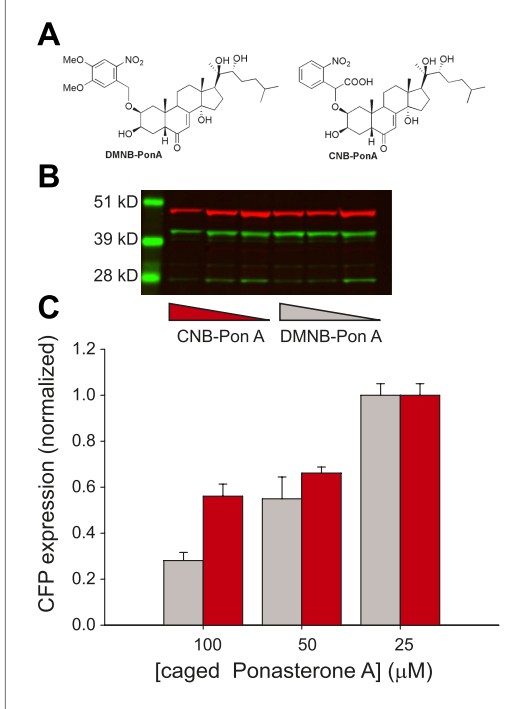

**Figure 5**. Caged PA is an ecdysone receptor antagonist. (**A**) Molecular structures of the two forms of caged PA: DMNB-PonA and CNB-PonA. (**B**) Dose dependent competition between caged PA and unmodified PA. The competition experiment consists of an overnight incubation in various doses of caged PA (CNB PonA = red triangle; DMNB-PonA = gray triangle), followed by incubation with un-modified PA. The expression is quantified by Western blot. The induced CFP-SKL marker is the lower green band (24 kD), the constitutively expressed MS2-YFP is the upper green band (45 kD), and the α-tubulin is the red band (50 kD). (**C**) Quantification of caged PA inhibition (gray = DMNB-PonA; red = CNB-PonA).

The following figure supplements are available for figure 5:

**Figure supplement 1**. Caged-PA can be switched from inactive to active with UV photolysis.

**Figure supplement 2**. Caged-PA competes with unmodified PA.

into distinct processes, which could be measured independently in the microscope. Although a number of studies have inferred transcriptional bursting by measuring events which are downstream of transcription, such as cellular RNA or protein production, (*Blake et al., 2003*; *Raser and O'Shea, 2004*; *Becskei et al., 2005*; *Bar-Even et al., 2006*; *Newman et al., 2006*; *Raj et al., 2006*; *Zenklusen et al., 2008*; *Singh et al., 2010*; *Batenchuk et al., 2011*; *Skupsky et al., 2010*; *So et al., 2011*; *Suter et al., 2011*), we are able to unambiguously resolve the properties of individual bursts and relate these measured kinetic steps to steroid abundance. For this ecdysteroid-responsive reporter gene in mammalian cells, the time between on-states decreased at higher doses of PA. The duration of the active state and the transcription rate from the active state were invariant. However, the 'on' times and 'off' times were both exponentially distributed (*Figure 2G,H*), which suggested that both processes were determined by a single rate-limiting step. The simplest biochemical model to explain this result is that the state of the gene is determined entirely by the on rate and off rate of the active ligand-receptor complex from the response element. The corollary to this model is that the actual transcription rate (*c*) from the active state is determined by factors other than the response element. For example the ability of *trans*-acting factors to access *cis*-acting sequences in the core promoter may determine the firing rate from the active state. In agreement with this supposition, the dynamic frequency modulation by the steroid was invariant with the insertion site, but the transcription rate varied over fivefold with insertion site. In this way, genetic regulation can be conceptualized as the superposition of individual parts, with distal enhancers and response elements controlling frequency of activation and proximal promoters determining the firing rate. Future experiments will be able to directly test this hypothesis using further development of the single-molecule approaches described here.

This simple model must address the fact that the exchange rate of SRs on chromatin has been observed to be extremely dynamic. Fluorescence

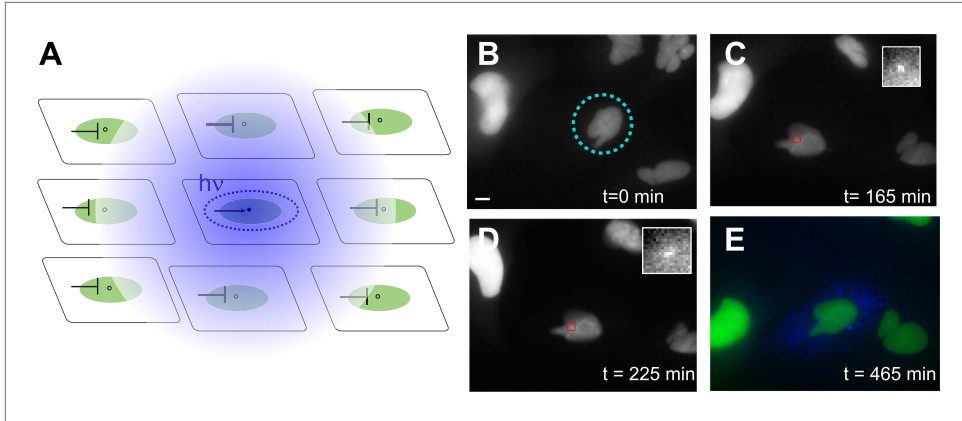

**Figure 6**. Photoactivation of single genes in vivo relates agonist kinetics to transcription dynamics. (**A**) Schematic uncaging experiment in tissue culture. The target cell is photolyzed with a laser that has the physical dimensions of a single nucleus (dotted circle). The transcription site in that cell becomes active, but transcription sites in neighboring cells are still repressed due to the antagonistic effects of caged PA. Thus, even though PA will diffuse through the membrane (indicated by hazy blue circle), the neighboring cells will not activate. (**B–E**) Time-lapse images of expression for an uncaged cell (ligand = DMNB-PonA). The uncaging spot is designated by a dotted blue circle. The transcription site in the activated cell is evident at 165 min and persists for 60 min. By 465 min, the peroxisomes are visible. Each frame is a *z*-stack of 30 images taken at 0.5 µM increments. Frame interval = 15 min. Images are maximum projected *z*-stacks. To maintain viability, only the YFP channel is imaged during acquisition, with CFP imaging utilized as an endpoint assay. Scale bar = 4 µm.

photobleaching recovery measurements on tandem gene arrays in single cells record dwell times on the order of seconds to minutes (*Darzacq et al., 2009*), while the duration of the active transcriptional state in this study is on the order of an hour. One possibility is that SR binding is actually much longer, and FRAP data on arrays is dominated by non-specific interactions and is not capable of detecting the small percentage of receptors which might be stably bound. Another possibility is that SR binding is in fact short-lived but initiates a cascade of events involved in activation of the gene (*Herschlag and Johnson, 1993*; *Hager et al., 2006*; *Metivier et al., 2006*). In support of this latter view, we observe that there was a delay between photoactivation of the ligand and the appearance of the first transcription sites (*Figure 6*). This delay was comparable in size to the average duration of the off period. In contrast, a non-chromatinized or weakly chromatinized template showed extremely rapid induction (<20 min). It is plausible that chromatin remodeling is inefficient at this reporter gene, possibly because the promoter lacks *cis*-acting elements which might aid in a more rapid chromatin remodeling response. Taken together, these results support the notion that opening of chromatin interposes a delay between the docking of active ligand-receptor complex on the response element and the eventual synthesis of pre-mRNA (*Janicki et al., 2004*).

The synergy between DNA binding factors and chromatin modifying proteins is essential for control of gene expression (*Segal and Widom, 2009*), and the role of chromatin in a bimodal model of gene regulation has been proposed previously

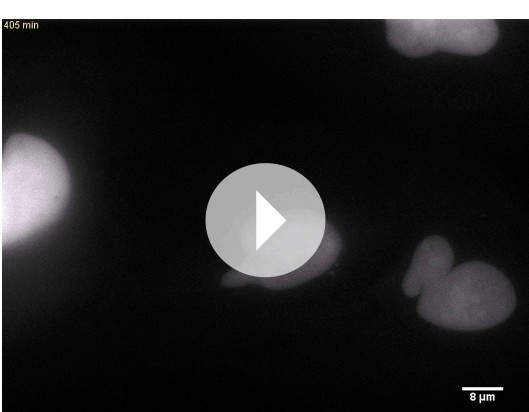

**Video 8**. Time-lapse sequence of reporter gene uncaging. The nascent transcription site is visualized as the coalescence of MS2-YFP coat protein on newly synthesized pre-mRNA. Cells were incubated in 100 µM DMNB-PA overnight. The media were then removed, the cells rinsed 3 × with fresh media without DMNB-PA, and then irradiated with 40 pulses at 100 Hz. Each frame is the maximum projection of 28 z-steps acquired at 0.5 µm intervals. Exposure time = 200 ms. Frame interval: 15 min. Total duration: 8 hr. T = 37°C.

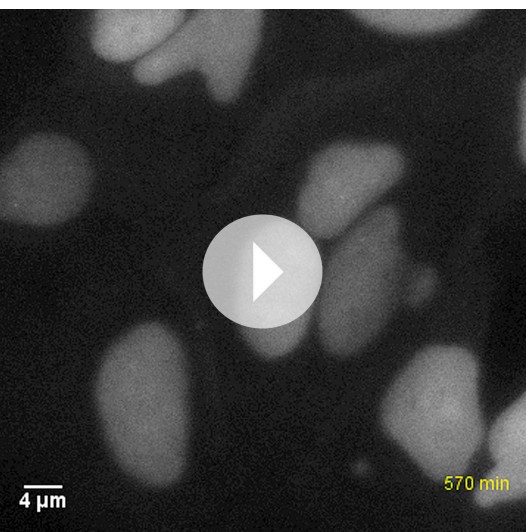

**Video 9**. Time-lapse sequence of reporter gene uncaging. The nascent transcription site is visualized as the coalescence of MS2-YFP coat protein on newly synthesized pre-mRNA. Cells were incubated in 100 μM DMNB-PA overnight. The media were then removed, the cells rinsed 3 × with fresh media without DMNB-PA, and then irradiated with 40 pulses at 100 Hz. In this instance, the uncaged cell proceeded through mitosis, and only one of the daughter cells shows activation. Each frame is the maximum projection of 28 z-steps acquired at 0.5 μm intervals. Exposure time = 200 ms. Frame interval: 30 min. Total duration: 15 hr. T = 37°.

(*Archer et al., 1992*; *Di Croce et al., 1999*). Moreover, recent evidence points strongly toward chromatin as directly controlling the duration of the on and off states. Histone methylation has been implicated in the transmission of transcriptional frequency between mother and daughter cells (*Muramoto et al., 2010*). Manipulation of histone acetylation changes transcriptional cycle timing (*Harper et al., 2011*; *Suter et al., 2011*). Histone deacetylases and nucleosome-remodeling complexes are cyclically recruited during transcriptional activation of an estrogen-responsive gene (*Métivier et al., 2003*). Importantly, FRAP studies on histones and genome-wide pulse-chase measurements on histones suggest a timescale of turnover (~hour) that is more consistent with the time scales of transcriptional activity observed in this study (*Kimura and Cook, 2001*; *Dion et al., 2007*; *Deal et al., 2010*). The bimodal 'telegraph' model is likely a simplification of the number of biological steps involved in gene activation, but the quantitative agreement between this model and our data suggests that at steady state, the requirements for re-activating a gene that has cycled off may depend on only a few rate-limiting steps.

In summary, we have described a comprehensive approach for unraveling the dynamic behavior of genetic control in single cells. By using a combination of nascent RNA visualization, single molecule microscopy, computational modeling, and light activation, we were able to construct a quantitative model of steroid-receptor mediated activation in which SRs control the frequency of the transition to an active transcriptional state in a dose-dependent manner. This model is based on a stochastic, kinetic description that reflects the molecular events occurring in single-cells.

## Materials and methods

### Construction of ponasterone responsive cell lines

The chimeric ecdysone receptor (pERV3, Stratagene, Agilent, Santa Clara, CA) was introduced into U2-OS cells by nucleofection (Amaxa, Lonza, Allendale, NJ), and stable insertions were selected by screening for G418 resistance followed by isolation of single colonies. Response to ecdysteroids was assayed using a luciferase reporter (pEGSH, Stratagene) transiently transfected into resulting cell lines. The most responsive clone was then used for all subsequent cell line derivation.

The reporter gene (*Figure 1A*) was cloned into the pHAGE lentiviral vector backbone described elsewhere (*Mostoslavsky et al., 2006*). Viral particles were produced from 293-T packaging cells in two 15 cm plates. The supernatant was harvested for three consecutive days. The collected supernatant was spun at max speed in a clinical centrifuge to pellet large debris and then filtered through a 0.45 μm filter before transfer to a sterile ultracentrifuge tube SW28. Viral particles were pelleted by spinning at 100,000 × *g* for 1.5 hr at 4°C. The entire collected supernatant was used to infect one dish of 5000 cells, resulting in infection efficiencies of ~90%.

The MS2 coat protein was also cloned into the same pHAGE backbone. The coat protein is driven off the human ubiquitin C promoter, contains a nuclear localization sequence and was described previously (*Fusco et al., 2003*). Lentivirus collection and infection were carried out according to the above protocol.

Unless otherwise stated, U2-OS cells were cultured in low-glucose DMEM (Gibco, Life Technologies, Grand Island, NY) supplemented with 10% FBS and 1% penicillin/streptomycin.

## Single-molecule RNA fluorescence in situ hybridization

Reporter RNA was visualized by hybridization of a 20-mer ssDNA oligo labeled with Cy3 at both ends to the MS2 repeats. Therefore, each 20-mer contains two dye molecules. The protocol for RNA FISH has been described elsewhere (*Femino et al., 1998*) and is used here with slight modification. Briefly, the coverslips with adherent U2-OS cells are washed three times with Hanks balanced salt solution and once with PBS followed by fixation with 4% paraformaldehyde for 10 min at room temperature. The cells were washed twice with PBS for 5 min each and permeabilized with 0.5% Triton X-100 in PBS for 10 min. The cells were washed with PBS and incubated twice in 10% formamide in 2 × SSC for 5 min before the hybridization is started.

Hybridization was carried out for 4 hr at 37°C in 10% formamide and 2 × SSC in the presence of competitor DNA and BSA. The probe concentration was 0.1–1 μM. After hybridization, the cells were washed twice in 10% formamide in 2 × SSC for 20 min at 37° and two times 1 hr in PBS at room temperature. The nucleus was stained for 10 s with 1 ml of 0.5 μg/ml DAPI followed by 2 × wash in PBS for 5 min. Coverslips were mounted using ProLong Gold antifade reagent (Invitrogen, Life Technologies, Grand Island, NY) and imaged 24 hr later.

## FISH image acquisition and analysis

Wide-field fluorescence images were obtained on a BX61 epi-fluorescence microscope (Olympus, Center Valley, PA) using a PlanApo 60 ×, 1.4 NA oil-immersion objective. The microscope is equipped with an X-Cite 120 PC light source (EXFO, Mississauga, Canada) for illumination, Uniblitz shutters (Vincent Associates, Rochester, NY), filter sets 31000 (DAPI), 41007a (Cy3) and 41008 (Cy5) (Chroma Technologies, Brattleboro, VT) and a MS-2000 Microscope Stage Controller (Applied Scientific Instrumentation, Eugene, OR). Digital images were acquired with a CoolSNAP HQ camera (Photometrix, Tucson, AZ) controlled by Metamorph 7.6.3 software as a 3D stack of 18 images at 0.3 μm steps with a 1 × 1 binning. Exposure times were 500 ms for Cy3 and Cy5 and 30 ms for DAPI. The fields for imaging are selected in the DAPI channel to minimize bias.

All image analysis was performed using custom-written software in IDL on maximum intensity projections of three-dimensional image stacks. Spot detection using a Gaussian mask (*Thompson et al., 2002*) was carried out using the Localize software package described previously (*Trcek et al., 2012*). Nucleus segmentation was achieved by applying an intensity threshold to the maximum intensity projection of the grayscale image. The threshold was adjusted manually to ensure accurate masking. The algorithm detected and removed nuclei that are either too small or not fully contained within the image in order to generate a binary mask. Cells were segmented using intensity thresholding in combination with the watershed operator.

## Western blot

We used the following primary antibodies: β-tubulin (Rockland, Gilbertsville, PA) and GFP (Roche, Branchburg, NJ), and secondary antibodies: donkey anti-rabbit conjugated to IRDye 800 (Rockland) and donkey anti-mouse conjugated to Alexa Fluor 680 (Invitrogen A21102). We washed cells in ice-cold PBS and lysed them at room temperature for 2 min in 1 ml lysis buffer per 10-cm dish (50 mM Tris-HCl (pH 8.0), 50 mM NaCl, 1% NP40, 5 mM DTT, 1 mM PMSF and half a mini tablet protease inhibitor). We spun the lysate 15 min at 14,000 × *g* at 4°C and loaded the supernatant on a Nupage 4–12% bis-tris gel using MOPS running buffer (Invitrogen). After transfer on a nitrocellulose membrane in Nupage transfer buffer (25 V for 1.5 hr), we blocked nonspecific interactions by incubating the blot overnight at 4°C in PBS supplemented with 5% nonfat dry milk and 1% BSA. After that, we rinsed the membrane and incubated it for 1 hr with the primary antibodies in PBS supplemented with 1% BSA (dilutions: 1:2500 mouse anti–Actb; 1:5,000 mouse anti–β-tubulin). We then washed the blot five times 10 min in PBS with 0.3% Tween-20 before incubation for 30 min with the secondary antibody (1:10,000 in PBS with 1% BSA). We then washed the membrane five times in PBS with 0.3% Tween-20, before exposure on an Odyssey infrared imaging system (two-color detection). Band intensities were quantified using custom software in IDL.

## Ecdysteroid preparation and analysis

Ecdysone and Ponasterone A were purchased from AG Scientific (San Diego, CA). The purity of ponasterone A was determined to be ~75% by analytical HPLC. All other reagents and solvents were purchased from Sigma-Aldrich (St. Louis, MO). Silica gel 60 (40 mm, Fischer) was employed for column chromatography. DMNB-Ecdysone and 2-bromo-2-(2-nitrophenyl)acetic acid were synthesized by

reported method (*Chang et al., 1995*; *Lin et al., 2002*). 1D and 2D NMR spectra were recorded on a Bruker DRX-500. High-resolution mass spectra were obtained at the Mass Spectrometry Facility, the University of North Carolina at Chapel Hill, Chapel Hill, NC.

## Preparation of DMNB-ponasterone A

A suspension of Ponasterone A (10 mg, 21 µmol) and dibutyltin oxide (6.7 mg, 27 µmol) in anhydrous methanol (3 ml) was heated to reflux for 3 hr under argon. After the solvent was removed under reduced pressure, the residue was subsequently azeotroped with anhydrous benzene (3 × 2 ml). The resulting stannylene acetal was further dried in vacuo for 2 hr before addition of 3 Å molecular sieves (50 mg), CsF (13 mg, 83 µmol), 1-bromomethyl-4,5-dimethoxy-2-nitrobenzene (11 mg, 38 µmol), and anhydrous DMF (1 ml). After the reaction mixture was stirred at room temperature overnight, the solvent was evaporated under reduced pressure. The resulting residue was purified by silica gel column chromatography (chloroform/methanol: 20:1) to afford caged-product as an off-white solid. DMNB-ponasterone A needs further purification by HPLC due to the impurity in the purchased ponasterone A (8 mg, 57%) (retention time 46.9 min on a Prevail C18 5 µ column 250 mm × 22 mm monitored at 242 nm; a 30 min linear gradient from 95% A [water] to 50% B [acetonitrile], followed by 50% B for 5 min with the flow rate of 8 ml/min). The purity of the product was determined to be >99% by analytical HPLC (retention time 43.4 min on an Apollo C18 5 µ column 250 mm × 4.6 mm, monitored at 242 nm; a 15 min linear gradient from 95% A [water] to 50% B [acetonitrile], followed by 50% B for 5 min and a 15 min linear gradient from 50% A [water] to 95% B [acetonitrile] with the flow rate of 1 ml/min). $^1$HNMR (500 MHz, CD$_3$OD): δ 0.91 (s, 3H), 0.94 (s, 3H), 0.95 (s, 3H), 1.01 (s, 3H), 1.20 (s, 3H), 1.2 (m, 2H), 1.5 −2.1 (m, 15H), 2.44 (m, 2H), 3.08 (m, 1H), 3.11 (m, 1H), 3.70 (d, $J$ = 11.5 Hz, 1H), 3.93 (s, 3H), 3.99 (s, 3H), 4.29 (s, 1H), 4.92 (d, $J$ = 15.0 Hz, 1H), 5.04 (d, $J$ = 15.0 Hz, 1H), 5.64 (s, 1H), 7.45 (s, 1H), 7.72 (s, 1H). $^{13}$CNMR (125 MHz, CD$_3$OD): δ 16.6, 19.6, 20.1, 21.4, 22.0, 22.9, 27.8, 29.0, 30.3, 31.1, 31.3, 33.2, 33.7, 36.3, 37.9, 47.2, 48.2, 49.0, 50.6, 55.4, 55.5, 64.2, 66.9, 76.2, 76.4, 76.6, 83.8, 107.7, 110.5, 120.8, 130.1, 139.7, 147.9, 153.6, 166.5, 204.8. HRMS (ESI$^+$) calculated for $C_{36}H_{53}NO_{10}Cs^+$: 792.2724 Found: 792.2716 (−1.0 ppm).

## Preparation of CNB-ponasterone A/ecdysone

CsF (13 mg, 83 µmol) was added to a solution of the stannylene acetal of ponasterone A/Ecdysone (21 µmol), 3 Å molecular sieves (50 mg), 2-bromo-2-(2-nitrophenyl)acetic acid (11 mg, 38 µmol) in anhydrous DMF (1 ml). After the reaction mixture was stirred at room temperature overnight, the solvent was filtered to remove 3 Å molecular sieves and evaporated under reduced pressure. The resulting residue was dissolved in DMSO and purified by HPLC.

## CNB-ponasterone A

White solid (7 mg, 52%) (retention time 15.1 min on a Prevail C18 5 µ column 250 mm × 4.6 mm, monitored at 242 nm a 40 min linear gradient from 30% A [water] to 95% B [acetonitrile], with the flow rate of 8 ml/min). $^1$HNMR (500 MHz, CD$_3$OD): δ 0.91 (s, 3H), 0.94 (s, 3H), 0.95 (s, 3H), 1.01 (s, 3H), 1.20 (s, 3H), 1.2 (m, 2H), 1.5 −2.1 (m, 15H), 2.44 (m, 2H), 3.08 (m, 1H), 3.11 (m, 1H), 3.60 (d, $J$ = 11.5 Hz, 1H), 4.15 (s, 1H), 4.92 (m, 1H), 5.72 (s, 1H), 7.46 (t, $J$ = 9.5 Hz, 1H), 7.62 (t, $J$ = 9.5 Hz, 1H), 7.75 (d, $J$ = 9.5 Hz, 1H), 7.93 (d, $J$ = 10.0 Hz, 1H). HRMS (ESI$^+$) calculated for $C_{34}H_{49}NO_8Na^+$ (M–CO$_2$): 622.3356 Found: 622.3355 (−1.0 ppm).

## CNB-ecdysone

White solid (9 mg, 65%) (retention time 34.5 min on a Prevail C18 5 µ column 250 mm × 22 mm monitored at 242 nm; a 30 min linear gradient from 95% A [water] to 50% B [acetonitrile], followed by 50% B for 5 min and a 15 min linear gradient from 50% A [water] to 95% B [acetonitrile] with the flow rate of 8 ml/min). $^1$HNMR (500 MHz, CD$_3$OD): δ 0.80 (s, 3H), 0.89 (s, 3H), 1.14 (s, 3H), 1.21 (s, 6H), 1.22–1.98 (m, 17H), 2.39 (m, 2H), 3.11 (m, 1H), 3.60 (d, $J$ = 11.5 Hz, 1H), 4.15 (s, 1H), 4.92 (m, 1H), 5.72 (s, 1H), 7.46 (t, $J$ = 9.5 Hz, 1H), 7.61 (t, $J$ = 9.5 Hz, 1H), 7.74 (d, $J$ = 9.5 Hz, 1H), 7.92 (d, $J$ = 10.5 Hz, 1H). HRMS (ESI$^+$) calculated for $C_{34}H_{49}NO_9Cs^+$ (M–CO$_2$): 748.2461 Found: 748.2496 (+4.6 ppm).

## Fitting to the random telegraph model

The number of mRNA/cell was binned into histograms and fit with the analytical solution to the random telegraph model (*Raj et al., 2006*) using Mathematica (Wolfram Research, Champaign, IL). The probability was summed over the bin size to generate the proper normalization. Unless otherwise indicated, there are three fitting parameters: *a/d*, *b/d*, *c/d*, corresponding to the normalized gene activation rate, gene inactivation rate, and initiation rate, respectively.

## Live-cell microscopy

Live-cell fluorescence images were obtained on a BX61 epi-fluorescence microscope (Olympus) modified for simultaneous UV photolysis. The system consists of an X-Cite 120 PC light source (EXFO, Mississauga, Canada) for imaging, an Explorer 339 nm pulsed laser for photolysis (Newport, Santa Clara, CA), and ASI shutters and automated stage (Applied Scientific Instrumentation). The laser is capable of operating from single-shot to 1 kHz at 120 μJ/pulse. All data was acquired at 37°C using a Delta T stage incubator and objective heater (Bioptechs, Butler, PA). The UV and visible excitation were focused through a U-Apo 40 × 1.35 NA oil lens, and the emission was collected through the same objective. Excitation, emission, and dichroic optics were custom designed for simultaneous UV and visible excitation (Chroma). The microscope was controlled by IPLab (BioVision, Exton, PA), and images were acquired on an Orca R2 CCD (Hamamatsu, Hamamatsu, Japan).

Time-lapse images of active transcription were analyzed using previously described methods (*Larson et al., 2011*). First, nuclei are segmented and individual cell videos are separated out from the raw data. The intermediate which results is a time-lapse image stack where the fluorescent nucleus is always centered in the frame (*Video 5*). Next, the diffraction-limited spots are fit using a Gaussian mask algorithm (*Thompson et al., 2002*) to generate a list of positions. The complete time-dependent trajectories were then constructed using a tracking algorithm (*Crocker and Grier, 1996*). We used a Hidden Markov Model fitting algorithm taken directly from Lee et al. to analyze the resulting intensity traces and extract the duration of active and inactive states (*Lee, 2009*).

## Acknowledgements

The authors gratefully acknowledge the Transcription Imaging Group at the Janelia Farm Research Campus of the HHMI and helpful discussions with Gordon Hager and Aviv Bergman. Jeff Chao, Timothee Lionnet, Antoine Coulon, and Stoney Simons provided critical feedback on the manuscript. Xavier Darzacq and Yaron Shav-Tal constructed the original cell lines and plasmids.

## Additional information

### Competing interests

RHS: Reviewing editor, *eLife*. The other authors declare that no competing interests exist.

### Funding

| Funder | Grant reference number | Author |
| --- | --- | --- |
| National Institutes of Health | GM 86217 | Robert H Singer |
| National Cancer Institute | Intramural Research Program of the NIH | Daniel R Larson |

The funders had no role in study design, data collection and interpretation, or the decision to submit the work for publication.

### Author contributions

DRL, Conception and design, Acquisition of data, Analysis and interpretation of data, Drafting or revising the article, Contributed unpublished essential data or reagents; CF, Acquisition of data, Analysis and interpretation of data; LS, XM, Acquisition of data, Contributed unpublished essential data or reagents; DSL, Conception and design, Drafting or revising the article; RHS, Conception and design, Analysis and interpretation of data, Drafting or revising the article

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
