## [Decision Letter]

Thank you for choosing to send your work, “Light-Activated Receptor-Mediated Transcription in Single Cells”, for consideration at *eLife*. The Editor-in-Chief, a Senior Editor, a member of the Board of Reviewing Editors, and two expert reviewers have assessed your work.

We have some interest in the publication of this work, but there are several key issues that need to be addressed before we can proceed further. We therefore ask whether you would be willing to respond to the specific points below.

1) The reviewers appreciated the novelty of the use of a next-generation system for the analysis of steroid receptor-mediated activation at the single cell/single reporter/single molecule level. It was also felt, however, that in spite of the technical novelty the main conclusions of this work mostly confirmed previous observations indicating that the on-off states are digital in nature. We therefore ask if the authors could clarify the novelty of their findings in the context of prior knowledge.

2) There is missing information that is needed to understand the results. Several examples are as follows:

a. Figure 2. It is necessary to show the data with 0 micromolar PA. This is the baseline control.

b. Figure 4. The authors conclude that local chromosomal environment affects initiation rates by comparing different clonal cell lines that accrue different amounts of total labeled construct mRNA. Were analyses, such as those in Figure 2, carried out for each clonal cell line to confirm for each the dose-dependent off-time and dose-independent on-time? (It is not apparent if Figure 2 shows data from one clonal cell line or the averages of all three clonal lines.)

c. The cells in Figures 3 and 6 look sick. In particular, the nuclear blebbing and irregularly shaped nuclei are indicative of apoptosis. This needs to be examined carefully as it could alter the interpretation of the data.

d. Exposing cells, and in particular nuclei, to 349nm light can result in DNA damage. Labeling with **γ**-H2A.X must be done +/- uncaging to be sure that the observed results are not influenced by DNA damage.

e. Figure 6. It appears that the authors presume that the PA that is released upon UV treatment exists only transiently because it becomes blocked by an excess of caged PA. Is this what the authors mean to suggest? If so, can they prove this point?

f. Can the authors confirm that there is only one transcription site per cell? Is this single transcription site stably maintained through multiple generations?

3) There is also an issue with the Random Telegraph model in Figure 4. In Figure 4, is the x-axis equal to time? If so, wouldn't (a) and (b) be times rather than rates? Also, would the fraction of time in the active state be equal to (b)/(b)+(a)? Please clarify the formulas as well as the use of the terms “rate” vs “time” in this section.

4) It is unclear why the first transcription sites are only visible several hours after induction. Opening of chromatin of a single copy gene should not impose such a large delay between docking of the active ligand-receptor complex and subsequent RNA synthesis. Is this a result of the complexity of the reporter system? How well does this reflect the induction of endogenous steroid regulated genes? This can be easily tested by in situ hybridization.

5) Figure 6. The authors assert “a single on-period in 70% of the cells which activated.” Were there multiple on-periods in the other 30% of activated cells? Authors should show statistical analyses of the probabilities of single or multiple cycles occurring from a single transient hormone dose. Additionally the average, maximum, and minimum on-periods should be calculated and presented for this experiment and compared to steady state hormone dosing assays.

---

## [Author Response]

*1) The reviewers appreciated the novelty of the use of a next-generation system for the analysis of steroid receptor-mediated activation at the single cell/single reporter/single molecule level. It was also felt, however, that, in spite of the technical novelty, the main conclusions of this work mostly confirmed previous observations indicating that the on-off states are digital in nature. We therefore ask if the authors could clarify the novelty of their findings in the context of prior knowledge*.

We point out that it is likely that *all* metazoan genes are transcribed by bursts of transcription, meaning that short periods of RNA synthesis are interspersed by long periods of inactivity. The causes of transcriptional bursting are unknown, but this ubiquitous phenomenon has profound consequences for development, differentiation, and disease progression. Therefore, understanding the mechanisms of gene regulation in vivo requires understanding how *cis*- and *trans*-acting elements work in concert to modulate properties of the gene burst. With the exception of one other study that observed single gene bursting directly (Yunger, et al. (2010)), all other studies in mammalian cells are based on measuring protein distributions and inferring transcriptional activity. While such studies have been informative, they depend on numerous assumptions about RNA export, translation, RNA decay, proteolysis, etc. Such an approach would not have revealed the phenomena we observed in our work. In this study, we have shown that steroid receptors, which are present in all metazoans and are also the target of a billion dollar pharmaceutical industry, have a mechanism of action in vivo (frequency modulation) that is at odds with the standard pharmacological description of the last 60 years. This advance in basic biological understanding is concomitant with technical advances such as development of a true opto-genetic method for uncaging single genes and the ability to monitor single genes for hours in vivo. We believe this work establishes a template for understanding gene regulation at the cellular and molecular level. In the revised manuscript, we have made a more strenuous effort to emphasize the novelty of the approach and the findings.

*2) There is missing information that is needed to understand the results. Several examples are as follows*:

*a. Figure 2. It is necessary to show the data with 0 micromolar PA. This is the baseline control*.

We are unable to detect any RNA (Figure 4) or protein product (Figure 4) at 0 micromolar PonA, which is an indication of the tightness of the system. Under these conditions, we also do not see transcription sites, either by fixed-cell FISH or live-cell imaging.

*b.*
Figure 4*. The authors conclude that local chromosomal environment affects initiation rates by comparing different clonal cell lines that accrue different amounts of total labeled construct mRNA. Were analyses, such as those in*
Figure 2*, carried out for each clonal cell line to confirm for each the dose-dependent off-time and dose-independent on-time. (It is not apparent if*
Figure 2
*shows data from one clonal cell line or the averages of all three clonal lines.*)

The analysis in Figure 2 was done on a clonal cell line. We have clarified this in the text. In the original version, subsequent cell lines were only analyzed by FISH (Figure 4), demonstrating the insertion effects on transcription dynamics. We have since repeated the live-cell analysis for another clone and observed the same frequency modulation behavior.

*c. The cells in Figures 3 and 6 look sick. In particular, the nuclear blebbing and irregularly shaped nuclei are indicative of apoptosis. This needs to be examined carefully as it could alter the interpretation of the data*.

We have chosen different cells for the figures and videos where necessary so as to avoid any confusion on this matter. In addition, we have added data showing the export of RNA into the cytosol along with cell division to demonstrate that these cells are healthy, dividing cells (Video 7).

*d. Exposing cells, and in particular nuclei, to 349nm light can result in DNA damage. Labeling with*
**γ***-H2A.X must be done +/- uncaging to be sure that the observed results are not influenced by DNA damage*.

We share the reviewers’ concern about potential DNA damage. However, to see **γ**-H2A.X staining requires a dose of 60 J/cm^2^ UVA (Wischermann, et al. (2008)). Our dose is 2 J/cm^2^, which is an order of magnitude below that threshold. We have added that information to the main text and the Materials and methods. We have also included a time-series of cell division after uncaging (Video 9) that we think is a more functional phenotype for cell viability after UV exposure.

*e. Figure 6. It appears that the authors presume that the PA that is released upon UV treatment exists only transiently because it becomes blocked by an excess of caged PA. Is this what the authors mean to suggest? If so, can they prove this point*?

We have added an additional experiment to show that upon addition of caged PA to an actively transcribing sample, transcription sites disappear over time (Figure 5).

*f. Can the authors confirm that there is only one transcription site per cell? Is this single transcription site stably maintained through multiple generations*?

It is certainly possible that lentiviral infection would result in multiple insertions in different parts of the genome. We performed infection at MOI <1, resulting most often in 1 or 0 integrations per cell. However, in some of our subclones we do see multiple integrations. For the cell lines we sub-cloned and analyzed in Figures 2 and 4, we never observed more than one transcription site per cell, as indicated in the data we present in the paper. Since lentiviral infection is a standard technique that results in genomic integration, we have no reason to suspect that the reporter gene is not stable. Indeed, we can passage cells dozens of times without seeing any change in the microscope.

*3) There is also an issue with the Random Telegraph model in Figure 4. In Figure 4, is the x-axis equal to time? If so, wouldn't (a) and (b) be times rather than rates? Also, would the fraction of time in the active state be equal to (b)/(b)+(a)? Please clarify the formulas as well as the use of the terms “rate” vs. “time” in this section*.

We have clarified the text and the figure on these points. a, b, c are rates. 1/a, 1/b, 1/c are times.

*4) It is unclear why the first transcription sites are only visible several hours after induction. Opening of chromatin of a single copy gene should not impose such a large delay between docking of the active ligand-receptor complex and subsequent RNA synthesis. Is this a result of the complexity of the reporter system? How well does this reflect the induction of endogenous steroid regulated genes? This can be easily tested by in situ hybridization*.

Since there are no other published examples of steroid response following a brief burst of ligand (as we achieve with uncaging), there is no reference point to which we can compare our results. Most studies look at the time response to a step increase in ligand concentration, where there is continuous availability of the ligand. With that caveat in mind, we agree with the referees that the long time lag is intriguing, especially since there is essentially no time lag for a transfected reporter. These data suggest that chromatin remodeling is perhaps inefficient at the reporter. It is possible that these exogenous *cis*-acting elements in the reporter lack features that aid in the rapid chromatin remodeling response for an endogenous steroid responsive gene. We have added several sentences to the Discussion to address this point. Building a reporter from the bottom up that recapitulates the rapid steroid response after uncaging is likely to yield fundamental insight. Such experiments are in progress but are beyond the scope of this study.

As for endogenous steroid responsive genes, the comparison is difficult. To our knowledge, there are no genes in mammalian cells that respond to ecdysteroids.

*5) Figure 6. The authors assert “a single on-period in 70% of the cells which activated.” Were there multiple on-periods in the other 30% of activated cells? Authors should show statistical analyses of the probabilities of single or multiple cycles occurring from a single transient hormone dose. Additionally the average, maximum, and minimum on-periods should be calculated and presented for this experiment and compared to steady state hormone dosing assays*.

We have added the statistical quantification of this experiment and commented on the comparison in the Discussion.

References

Yunger S, et al, Single-allele analysis of transcription kinetics in living mammalian cells. Nat Meth, 2010. **7**(8): p. 631-633.

Wischermann K, et al, UVA radiation causes DNA strand breaks, chromosomal aberrations and tumorigenic transformation in HaCaT skin keratinocytes. Oncogene, 2008. **27**(31): p. 4269-4280.